# Species Discrimination of Three *Odontomachus* (Formicidae: Ponerinae) Species in Thailand Using Outline Morphometrics

**DOI:** 10.3390/insects13030287

**Published:** 2022-03-14

**Authors:** Yudthana Samung, Tanawat Chaiphongpachara, Jiraporn Ruangsittichai, Patchara Sriwichai, Anon Phayakkaphon, Weeyawat Jaitrong, Jean-Pierre Dujardin, Suchada Sumruayphol

**Affiliations:** 1Department of Medical Entomology, Faculty of Tropical Medicine, Mahidol University, Bangkok 10400, Thailand; yudthana.sam@mahidol.ac.th (Y.S.); jiraporn.rua@mahidol.ac.th (J.R.); patchara.sri@mahidol.ac.th (P.S.); anon.pha@mahidol.ac.th (A.P.); 2Department of Public Health and Health Promotion, College of Allied Health Science, Suan Sunandha Rajabhat University, Samut Songkhram 75000, Thailand; tanawat.ch@ssru.ac.th; 3Office of Natural Science Research, National Science Museum, Technopolis, Khlong 5, Khlong Luang, Pathum Thani 12120, Thailand; polyrhachis@yahoo.com; 4Institut de Recherche pour le Développement (IRD), UMR INTERTRYP IRD-CIRAD, University of Montpellier, F-34398 Montpellier, France

**Keywords:** outline morphometrics, *Odontomachus monticola*, *O. rixosus*, *O. simillimus*

## Abstract

**Simple Summary:**

Determination of species in the ant genus *Odontomachus*, which is a venomous group of ants, may require the use of highly trained entomologists. In Thailand, three species are very similar and difficult to distinguish: *O. monticola*, *O. rixosus,* and *O. simillimus*. In such a situation, a complementary technique not requiring highly specialized entomological knowledge is welcome. The geometric morphometric approach has proven to be this sort of tool, especially powerful for morphologically close or even cryptic species. In its most recent development, the geometric method uses the relative position of some anatomical landmarks. However, in worker ants these landmarks are few in number and can be difficult to assess without dissection. Here, therefore, we use the outline-based approach, an alternative geometric technique that has not yet been tested in ants. We show that the simple outline of the head contains a strong taxonomic signal, much stronger than the one obtained from the pronotum shape. The outline technique therefore represents a promising approach to aid in the determination of ant species.

**Abstract:**

All members of the ant genus *Odontomachus* Latreille, 1804 are venomous ants. Four species in this genus have been identified from Thailand: *Odontomachus latidens* Mayr, 1867; *O. monticola* Emery, 1892; *O. rixosus* Smith, 1757; and *O. simillimus* Smith, 1758. The three latter species are available and have been used for an outline morphometric study. They display similar morphology, which makes their distinction very difficult except for highly qualified individuals. A total of 80 worker specimens were studied, exploring the contour shapes of their head and pronotum as possible taxonomic characters. The size of each body part was estimated determining the contour perimeter, the values for which were largely overlapping between *O. rixosus* and *O. simillimus*; most *O. monticola* specimens exhibited a significantly larger size. In contrast to the size, each contour shape of the head or pronotum established *O. rixosus* as the most distinct species. An exploratory data analysis disclosed the higher taxonomic signal of the head contour relative to the pronotum one. The scores obtained for validated reclassification were much better for the head (99%) than for the pronotum (82%). This study supports outline morphometrics of the head as a promising approach to contribute to the morphological identification of ant species, at least for monomorphic workers.

## 1. Introduction

Biodiversity plays a vital role in maintaining the ecological balance on Earth. Ants are social insects and important members of ecosystems who act as ecosystem engineers because of their livelihood behaviors associated with below-ground processes that affect plants, microorganisms, and various soil organisms [1]. Approximately 15,000 ant species have been discovered so far worldwide and classified into 16 subfamilies and 296 genera [2]. In Thailand, 529 species in 109 genera within 10 subfamilies have been recently recorded [3]. The ant genus *Odontomachus* is one of the 10 genera of venomous ants [4,5,6]. Of them, six, i.e., *Formica*, *Tetramorium*, *Pachycondyla*, *Solenopsis*, *Hypoponera*, and *Odontomachus*, have been found in Thailand [3].

The ponerine ant genus *Odontomachus* Latreille, 1804, is assigned to the tribe Ponerini, subfamily Ponerinae, and family Formicidae [7,8]. This genus has so far been found in pantropical, pansubtropical, and temperate zones, containing 73 valid extant and three valid fossil species [9]. Workers of the *Odontomachus* genus are easy to identify from those of any other genus of the subfamily Ponerinae. They present a trap jaw, a part shared with the sister genus, *Anochetus* Mayr, 1861 [7]. The unusual *Odontomachus* trap mandibles and head shape are synapomorphic with those of *Anochetus*, but the genera are easily differentiated by the posterior part of the head. In *Odontomachus*, the nuchal carina is V-shaped along the median and the posterior surface of the head, displaying a pair of dark converging apophyseal lines. In *Anochetus*, the nuchal carina is continuously curved, and the posterior surface of the head lacks visible apophyseal lines. These genera also tend to differ in terms of size (*Anochetus* are generally smaller, though there is some overlap), propodeal teeth (absent in *Odontomachus* but usually present in *Anochetus*), and petiole shape (always coniform in *Odontomachus* but variable in *Anochetus*).

Four *Odontomachus* species have been reported from Thailand: *O. latidens* Mayr, 1867; *O. monticola* Emery, 1892; *O. rixosus* Smith, 1757; and *O. simillimus* Smith, 1758 [3]. Their external morphology is similar in general appearance, which makes species identification difficult. Most recent studies have proposed using DNA sequencing to make a distinction among these genera [8,10,11,12].

Only a few alternative or complementary techniques are available for entomologists to use for species recognition. Molecular techniques are currently the most accurate methods for such purposes [13]. These methods were recently used in the ant genera *Anochetus* and *Odontomachus* [12]; however, the relatively high cost of specialized laboratory equipment and the need for unexpired chemical reagents and well-trained staff could make their routine application challenging.

Outline morphometrics (OM) does not require any equipment other than optical and computer devices that are generally present in entomological laboratories. Moreover, this method can be successfully performed even by unskilled personnel [14]. Like the more recent geometric morphometric (GM) technique, OM allows separate analyses for size and shape coupled with visualization. In contrast to the GM method that is based on the relative position of anatomical landmarks (see an application on ants in Katzke et al. [15], Bagherian et al. [16], Seifert et al. [17], and Casadei-Ferreira et al. [18]), the OM method considers only the contours.

As a taxonomic tool, both GM and OM have proven to be useful in the morphological identification of arthropod species, including those that are closely related [14].

To the best of our knowledge, no studies to date have reported the use of OM in ant taxonomic identification. Therefore, this study used the OM method to compare the head and pronotum parts of the three very similar ant species in Thailand, *O. monticola*, *O. rixosus*, and *O. simillimus*. We suggest the OM method as a new alternative approach to help the morphological identification of worker ants.

## 2. Materials and Methods

### 2.1. Ant Specimens

We analyzed a total of 80 specimens of three ant species of the *Odontomachus* genus. All included specimens were taken from the Natural History Museum of the National Science Museum (Thailand) and were collected throughout Thailand between 2002 and 2018 (Appendix A). Only worker ants were used for this study. The workers of *Odontomachus* are monomorphic [2,19], i.e., their size variation is generally limited, with only a single distribution mode.

Using the contour of the head, we compared 25, 33, and 22 specimens of *O. monticola*, *O. rixosus*, and *O. simillimus*, respectively. For the pronotum, two specimens of *O. simillimus* were removed from the data because of an accidental deformation of their contour. The head and pronotum were photographed using a DS-Ri1 SIGHT digital camera attached to a Nikon AZ 100M stereomicroscope (Nikon Corp., Tokyo, Japan) under 1× magnification. A scale bar of 1 mm was set on every picture. To prevent fluctuations of the shooting angle, all ant samples were mounted in the same position and height using entomological pins.

Figure 1 shows the external morphology of *O. monticola*, *O. rixosus*, and *O. simillimus* workers. Figure 2 shows an example of the head and pronotum pictures used in OM analyses.

### 2.2. Morphometric Analyses

#### 2.2.1. Digitization and Digitization Error

The head and pronotum contours were determined using manually located pseudolandmarks without needing them to be equidistant or equal in number among the individuals. Although adopting the same starting point is not mandatory for the statistical method we used here, we digitized the contour targeting the same starting point. The contour used for the head was not completely anatomic; the starting and ending points were at the anterior of the head, did not exactly coincide, and were artificially joined by a straight line. This line was added by the digitizing program and was not an anatomical part (see arrow in Figure 2). However, it represented a very small portion of the external contour. This practice does not seem to affect the taxonomic signal between species [20,21].

The measurement error was estimated by the repeatability index (R) between two coordinate sets digitized from the same images by the same user. The R computing method used analysis of variance on repeated measurements; this is based on the classical variance estimators for size [22] and the Procrustes variance components for shape [23]. The measurement error [24] was one minus R.

#### 2.2.2. Size and Shape Data

A comparison of the head and pronotum sizes of the species was performed using the perimeter of the contour. The statistical significance of the size variation among species was based on non-parametric tests (1000 cycles) with a Bonferroni correction at a *p*-value of < 0.05.

For the head and pronotum, the contour shape was measured through the elliptic Fourier coefficients after normalization (NEF; for “Normalized Elliptic Fourier” coefficients) [25]. Coordinates of head and pronotum of three *Odontomachus* species are available in Appendix A.

#### 2.2.3. Exploratory Data Analysis

To provide direct evidence for the validity of the pre-established species proposition, we performed a principal component analysis based on the NEF describing each body part. The resulting morphospace (PC1, PC2) was illustrated by the factor map. A hierarchical analysis based on average link (unweighted pair group method with arithmetic mean: UPGMA) produced a tree where each item is an individual described by its seven first principal components.

#### 2.2.4. Discriminant Space

The shape divergence between the species was then quantified as Mahalanobis distances and illustrated by mapping individuals along the two discriminant factors (DF1 and DF2). To compute the Mahalanobis distances while avoiding the multidimensionality obstacle (overfitting), the scores of a limited number of principal components (PCs) of NEF were used instead of the original NEFs themselves. The criterion to determine the selected PC number was approximately the number of specimens in the smallest group divided by three (here 7PC).

The shape relatedness of individuals and species was illustrated by both the morphospace (PCA, PC1, and PC2) and by the discriminant space (DF1, DF2).

#### 2.2.5. Allometry

The separate analyses of size and shape do not imply complete independence between the two variable sets, i.e., their relationship is allometric. It is possible to estimate the allometric content of shape by regressing shape on size. As our objective was to distinguish species using shape (PCs of NEF), we limited the allometric study to the possible size effect on the shape-based discrimination. Therefore, the allometric effect was estimated using linear regression between the shape-derived discriminant factors DF1 and DF2 and the corresponding divergence of size (perimeter) [26].

#### 2.2.6. Mahalanobis-Based Classification

A validated classification procedure [27] was performed to test the accuracy of the individual assignment to their respective species. 

### 2.3. Software

Head and pronotum digitization, data processing, and analyses were performed using the CLIC package software version 99 [28,29]. Contour visualization and tree buildings used the XYOM program ([30], https://xyom.io/) (accessed on 22 December 2021). The UPGMA clustering tree was performed using the JMP software.

## 3. Results

### 3.1. Repeatability

Our head and pronotum digitization showed high and lower repeatability for the size (98.33% and 92.81, respectively) and shape (78.56% and 77.23, respectively), respectively.

### 3.2. Allometry

The allometric effect of the head size (perimeter) on head shape discrimination (DF1) was 19% for DF1 (*p* < 0.001, Figure 3) and 51% for DF2 (*p* < 0.001). The same analysis for the pronotum revealed a lower allometric effect (6%) for DF1 (*p* < 0.001, Figure 3) and for DF2 (46%, *p* < 0.001).

### 3.3. Size Variation among the Odontomachus Species

*O. monticola* exhibited the largest head (mean = 8.96 mm) and pronotum (mean = 5.55 mm) perimeter. The smallest head perimeter was observed in *O. rixosus* (mean = 7.17 mm), whereas the smallest pronotum was in *O. simillimus* (mean = 4.30 mm). All three species showed significant size differences for either the head or pronotum, except for *O. rixosus* and *O. simillimus,* who exhibited similar head size (Table 1, Figure 4).

### 3.4. Shape Variation among the Odontomachus Species

The mean shape superposition of the head and pronotum allowed us to highlight the areas where differences appeared between species. The posterior margin of the *O. simillimus* head contour was more curved than that of the other two species. The pronotum of *O. simillimus* also had a more curved shape than that of the other two species (Figure 5).

### 3.5. Morphospace and Exploratory Data Analyses

For the head contour, the principal components analyses showed almost non-overlapping clouds corresponding exactly to the initial grouping of species (Figure 6, top left). No such pattern was apparent for the pronotum, for which the three species largely overlapped (Figure 6, top right). For this reason, the exploratory data analysis was illustrated only for the head: it showed high agreement with pre-established hypotheses, except for three seemingly erratic individuals (4% of the total), all of them belonging to *O. monticola* (Figure 7). 

### 3.6. Discriminant Analysis

With three groups, the discriminant analyses produced two significant factors (DF1 and DF2). For the head and pronotum, the contribution of DF1 to the total shape variation was 55% (45% for the second factor) and 87% (13% for the second factor), respectively. The plot of these two DFs suggested a clear separation of the three species, more pronounced for the head than for the pronotum (Figure 6, bottom). Table 2 presents the statistical significance of Mahalanobis distances between species for head and pronotum shapes (pairwise Mahalanobis distances, 1000 runs; *p* < 0.001).

The non-overlapping distribution of the three species in the discriminant space and the higher head contour-derived discrimination were confirmed by the validated reclassification scores. For the head and pronotum, the total accuracy reached 99% and 82%, respectively (Table 3). 

## 4. Discussion

The results of this study revealed the efficacy of the OM approach in identifying three *Odontomachus* species with similar morphology, highlighting the particular interest of the head contour as a taxonomic character. 

Three common *Odontomachus* species are found in Thailand. *O. simillimus* is very similar in general appearance to *O. monticola,* as they share the broad head and dark body, but it is distinguishable from the latter by the presence of the erect setae on the pronotum, the first gastral tergite (lacking setae in the latter), and the sculpture on the lateral face of the petiole (smooth and shining in the latter). *O. risoxus* differs from *O. simillimus* and *O. monticola* in terms of the narrow and long head and yellowish body color. Moreover, the preapical tooth is clearly longer than broad in *O. rixosus* (as long as broad or shorter than broad in the other two).

Previous ant morphometric studies used linear distances between certain anatomical points while generally aiming to obtain a high number of measurements [31,32] to submit for univariate and sometimes multivariate statistical analyses [33]. These studies showed that certain morphological characteristics of the head or the pronotum could be used to help classify ant species.

Contrary to traditional morphometrics, modern morphometric developments use coordinates of points instead of linear distances between them [34]. Two main statistical methods are currently applied: GM, also called the landmarks-based approach [35], and OM, the outline-based approach [25,36].

The reason for the existence of different statistical treatments for the coordinates of anatomical points is the nature of the points. These points, called “landmarks”, are in the GM approach relocatable anatomical features with high homological content. The points used in the OM approach are today frequently called “pseudolandmarks”. In the OM approach, the homology of the morphological traits compared is expected to arise from the contour described by the pseudolandmarks and not from the pseudolandmarks themselves, whereas in the GM approach, the homology is expected for each landmark individually.

The GM method is applied when some anatomical landmarks are easily distinguishable from one individual to another, e.g., the cross-sectional areas on the wings of, for instance, mosquitoes [19,37], sandflies [38], and fruit flies [39]. Some recent morphometric studies in ants used the GM approach [14,16,17].

When the anatomical features of an organism present with few or no landmarks or would need entomological dissection [18], such as in the worker ants in this study or in non-winged insects, juvenile stages, or eggs, the OM approach can be used [13].

The main interest to use the coordinates of points (GM and OM) instead of the linear distances between them is that it becomes possible to capture the shape of an organ, visualize its changes between individuals or species, and measure shape and size divergence separately.

### 4.1. Size Variation among the Odontomachus Species

In our study, we estimated the head and pronotum size of three *Odontomachus* species by the perimeter of their observed contour. In the elliptic Fourier method developed here, the normalization of the Fourier coefficients was obtained by dividing the final coefficients by the semi-major axis of the first ellipse. This global size estimate is highly correlated to the perimeter of the observed contour, as well as to the square root of its internal area. We selected the perimeter due to its intuitive understanding.

Even if the head or pronotum size were apparently able to discriminate at least *O. monticula* from the other two species (Figure 4), we based the interspecies morphometric distinction only on shape. This strategy is due to the high environmental variance of size. Insect body size could be strongly influenced by environmental variables (e.g., breeding site availability, weather conditions, food availability during immature stages, etc.). It might lead to relevant overlap between the species, as here between *O. rixosus* and *O. simillimus*. Therefore, size is generally not recommended as a reliable taxonomic trait [26,40]. Indeed, in our material, the larger size of *O. monticola* could be due to its collecting site located at a higher altitude compared to that of the other two species (see Bergmann’s rule), so that smaller populations of this species cannot be excluded in other geographic areas. In contrast, larger populations of the other two species cannot be excluded in other geographic areas. Opposed to size variation, size divergence between species certainly displays genetic variation, at least in part. Therefore, the allometric content of our shape variables was not removed, being apparently higher for the head than the pronotum (Figure 3).

### 4.2. Shape Variation among the Odontomachus Species

The principal component analysis of the normalized Fourier coefficients, as illustrated by the morphospace (Figure 6, top), showed for the head a grouping of individuals in clear agreement with the three species as labeled by the National Science Museum (Thailand). Moreover, the hierarchical tree based on individual heads could separate the species with a minimum error (4%). Thus, the head confirmed the initial species identification of the specimens. The same approach applied to the pronotum could not provide such clear confirmation, suggesting that the taxonomic signal embedded in the contours of the ants varies according to the body parts.

The validated reclassification of the three *Odontomachus* ants, based on the Mahalanobis distances between them and using shape variables only, also showed higher total accuracy for the head contour (99%) than for the pronotum one (82%). The higher taxonomic accuracy of the head contour shape could be attributed in part to its higher allometric content (19%) relative to that of the pronotum (6%). This high accuracy of the head contour was obtained in spite of a slightly incomplete digitized contour (see arrow in Figure 2).

This study supports the OM method as a promising tool to help ant morphological identification, at least for species with monomorphic worker castes. It suggests the head contour as a much better taxonomic character than the pronotum one. The use of the contour of the head represents a special interest when the material to be identified cannot be dissected, as is the case for museum material.

## Figures and Tables

**Figure 1 insects-13-00287-f001:**
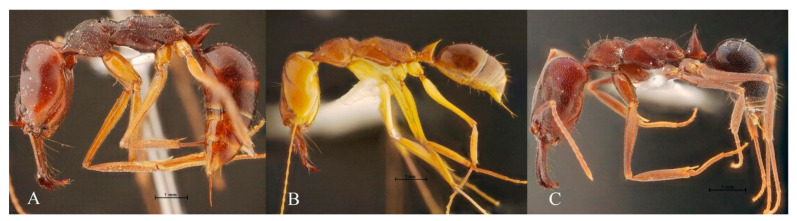
Morphology of workers *Odontomachus monticola* (**A**), *O. rixosus* (**B**), and *O. simillimus* (**C**).

**Figure 2 insects-13-00287-f002:**
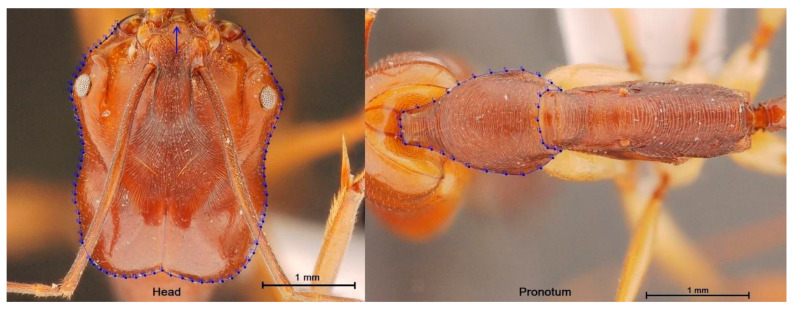
Pseudolandmarks (blue dots) describing the contour of the head (**left**) and of the pronotum (**right**). The arrow on the head shows the small part of the contour that could not receive valid pseudolandmarks.

**Figure 3 insects-13-00287-f003:**
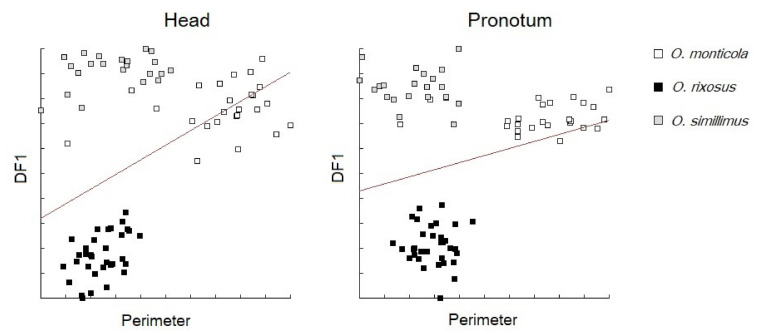
Linear regression lines between the first shape-based discriminant factor between three *Odontomachus* species and the corresponding size of head (**left**) and pronotum (**right**) contours.

**Figure 4 insects-13-00287-f004:**
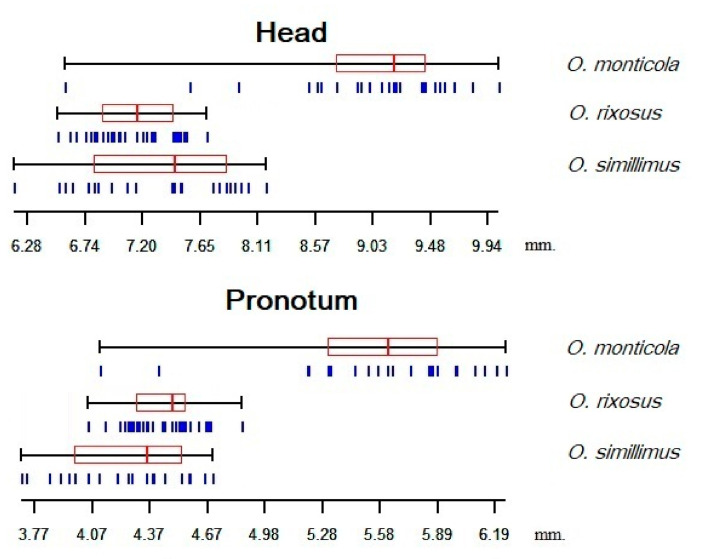
Boxplots illustrating size variations of head and pronotum contours (estimated by the perimeter). Each box represents each ant species and shows the group median that separates the 25th and 75th quartiles.

**Figure 5 insects-13-00287-f005:**
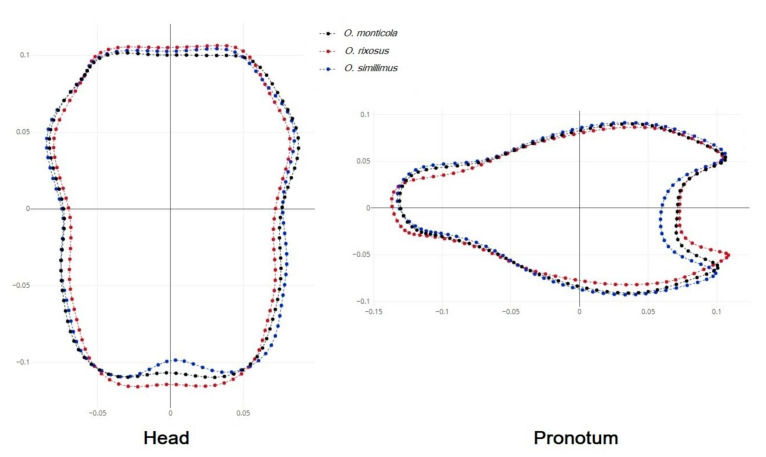
Superposition of mean shapes showing mean differences between *O. monticola* (black), *O. rixosus* (red), and *O. simillimus* (blue) for the head (**left**) and for the pronotum (**right**) contours.

**Figure 6 insects-13-00287-f006:**
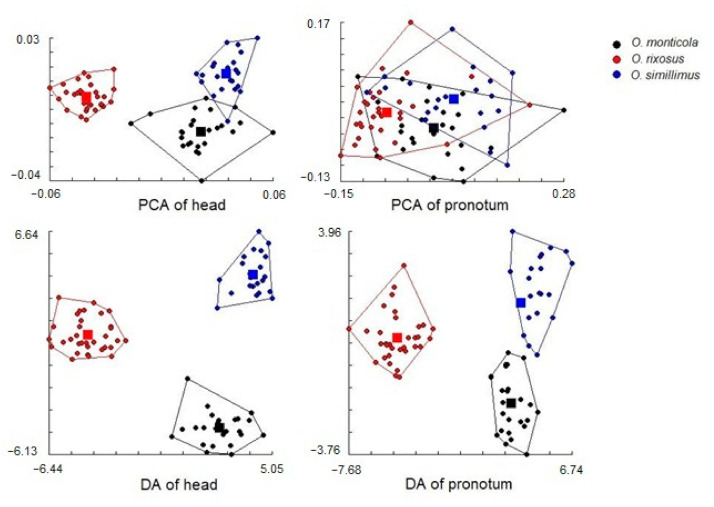
Top: Factor maps of the two first principal components (PC1 as the horizontal axis) of head (**left**) and pronotum (**right**) shape variables Bottom: Factor maps of the two discriminant factors (DF1 as the horizontal axis, DF2 as the vertical one) derived from the 7 first PC of shape variables of head (**left**) and pronotum (**right**). Each point represents an individual ant sample of either *O. monticola*, *O. simillimus*, or *O. rixosus*, and each *a posteriori* polygon corresponds to a different species. Squares represent mean values in each species.

**Figure 7 insects-13-00287-f007:**
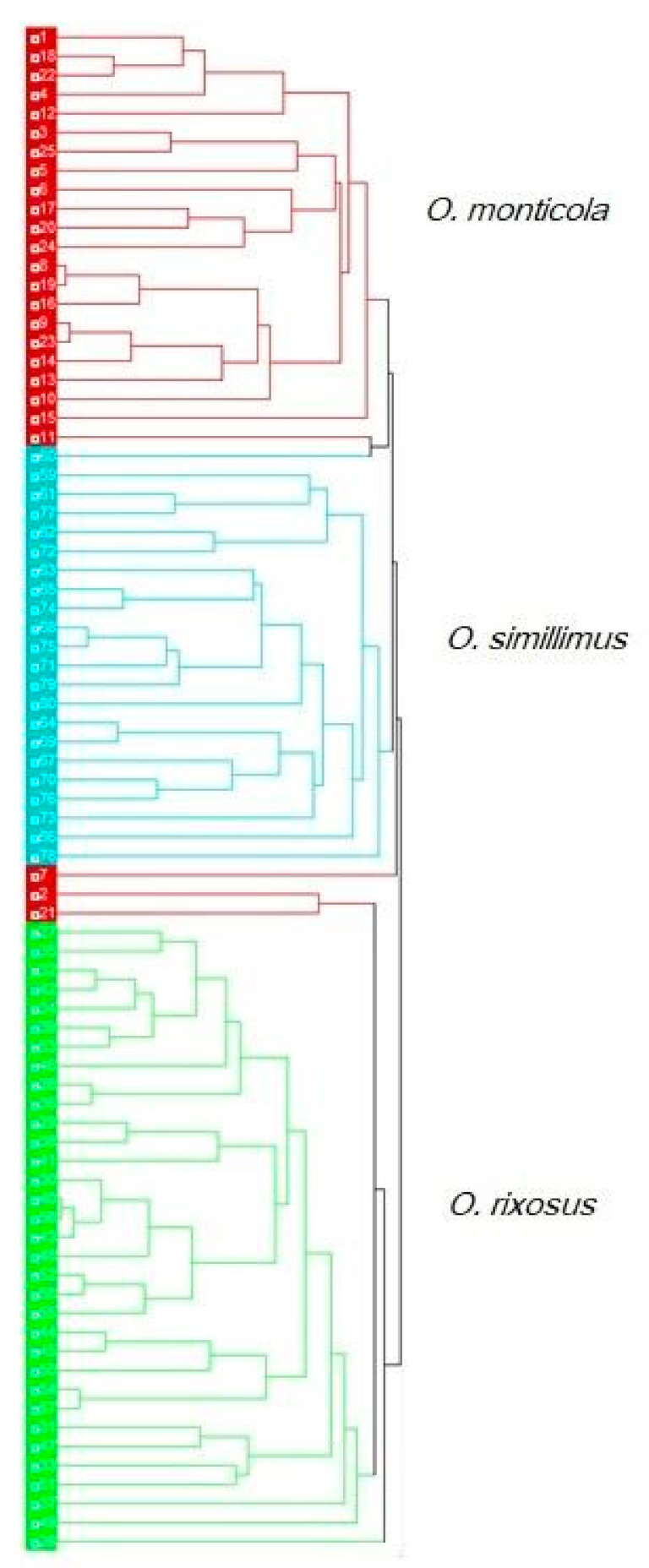
UPGMA hierarchical analysis generated from the 7 first principal components of shape variables describing the head of each individual.

**Table 1 insects-13-00287-t001:** Perimeter of head and pronotum contours of three *Odontomachus* species.

Species	*n*	Perimeter of Head				Perimeter of Pronotum			
		Mean (mm.)	Min (mm.)	Max (mm.)	SD	Mean (mm.)	Min (mm.)	Max (mm.)	SD
*O. monticola*	25	8.96 ^b^	6.67	9.94	0.71	5.55 ^c^	4.16	6.19	0.48
*O. rixosus*	33	7.17 ^a^	6.61	7.73	0.29	4.48 ^b^	4.09	4.86	0.18
*O. simillimus*	22	7.36 ^a^	6.28	8.18	0.55	4.30 ^a^	3.77	4.72	0.30

Different superscript letters indicate significant difference, *n*, sample size; Min, minimum; Max, maximum; SD, standard deviation.

**Table 2 insects-13-00287-t002:** Mahalanobis distances and statistical significance among three *Odontomachus* ants based on normalized Fourier coefficients.

Species	Head			Pronotum		
	*O. monticola*	*O. rixosus*	*O. simillimus*	*O. monticola*	*O. rixosus*	*O. simillimus*
*O. monticola*	0.00	-	-	0.00	-	-
*O. rixosus*	7.29 *	0.00	-	3.62 *	0.00	-
*O. simillimus*	6.89 *	7.11 *	0.00	1.78 *	4.24 *	0.00

Asterisks indicate significant differences at *p* < 0.001.

**Table 3 insects-13-00287-t003:** Percentage of correctly classified specimens based on cross-checked classification for head and pronotum.

Species	Head (*n* = 80)	Pronotum (*n* = 78)
*O. monticola*	96% (24/25)	76% (19/25)
*O. rixosus*	100% (33/33)	100% (33/33)
*O. simillimus*	100% (22/22)	60% (12/20)
Total	98.75% (79/80)	82% (64/78)

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
