# Peer review of "Species Discrimination of Three Odontomachus (Formicidae: Ponerinae) Species in Thailand Using Outline Morphometrics"

_insects, 2022, doi:10.3390/insects13030287_

Round 1
Reviewer 1 Report
This study employs outline morphometrics to differentiate three species of Odontomachus in Thailand. To do this, the authors explore outline data from head and pronotum. It is interesting because geometric morphometrics or outline morphometrics were not well explored in ant taxonomy (and so in phylogenies). Although I consider this study by Samung et al. welcome, I also feel that a few points need deeper discussion or consideration. For example, authors claim (mainly) that such an approach would help in ant identification of these three similar ant species. However, the authors did not provide any additional information about how or why someone would employ outline data to identify these species. Yet, did not mention any recommendation about how we could use outline studies to discriminate species (I mean, specimens replication, etc). Because this study is exploring such a morphometric approach for ants, I consider that providing recommendations for other people should be pursuit by authors and could be a good contribution of this study for other myrmecologists (and morphometric studies on insects, as well).
Further, is lacking in the Introduction/Discussion recent ant papers using geometric morphometrics (for example, Casadei-Ferreira et al. 2021: https://doi.org/10.1002/ECE3.7422) as well several studies in ant taxonomy using multivariate analyses from linear morphometric datasets (please, see studies by Csősz and Fisher, e.g., doi: 10.3897/zookeys.603.8271). Last year was published a global phylogeny of Odontomachus (and Anochetus), and should be cited in your paper (A global phylogenetic analysis of trap-jaw ants, Anochetus Mayr and Odontomachus Latreille (Hymenoptera: Formicidae: Ponerinae). Would be nice to see compared discussion exploring outline and traditional morphometric data in ant taxonomy or any kind of study discriminating ant species or populations.
- Please, check italic in through the text (there are so many scientific names without italic).
- L52 - what is missing? ‘and approximately 10000 species have been identified [2].’;
- indeed, from L52-56, text does not make sense;
-L59: 73 valid species (AntCat);
-L71-72: needs insert blank space in cited species names of Odontomachus; further, you should revise the entire text for such problem in the species names;
- Morphometric analyses - if the head capsule has bilateral symmetry, could you explain why outline the two sides of the head?
-L125: please provide additional information because it is not possible to understand ‘1-R’;
- The cluster analyses (for three objects) seems unnecessary: Figure 3 and Figure 6 already show the differences and similarities;
- Fig. 1 and Fig. 2 are not results but material and methods;
- Table 1 should be in supplementary material, not in main text; please also provide coordinates, so occurrence ant databases can benefit from your study;
- Fig. 4: you explain boxplot information in the legend, so remove the boxplot graph inside the figure;
- references Section is not exactly standardized; italics are lacking in many genus names or nominal species; further, there are double spaces in a few places needing correction/attention.
Reviewer 2 Report
In general, I think this is a relevant and interesting contribution to the field and I believe it is worthy of publication. However, I have some comments and concerns regarding a number of points which I will elaborate on below.
- Language check needed, some incorrect grammar and strange wording
- Line: 28: should be “in this genus”
- Line 34: Head and pronotum are not “organs” but rather “body parts”…
- Line 37: I’m not sure what is meant by “most external species”
- Line 48-50: Very strangely worded/arranged intro sentence, I would advise to rephrase this. I also advise against the use “small” as this is a very relative term. As a far as insects go, I would consider ants to be rather average size with some significant variation among species and genera.
- Line 53: this should probably say “recently recorded” instead of identified
- Line 60-62: please rephrase…
- Line 71: not “ants” but “species” …. “have been reported from”
- Line 166: no “the” in front of species list and “workers” instead of “worker ants”
- Line 243: should say „Odontomachus species“ …. „head and pronotum shapes“
- There are some issues with using the appropriate tense, Present Tense (not Past Tense) should be used in all parts of this manuscript where the authors talk about the study at hand!
For instance:
- Line 25: should be “show”
- Line 41: should say “this study supports”
- Line 312: should say “this study supports”
- Poisonous vs. venomous
First of all, this is the wrong word, a species that injects poison as it is the case with (some) ants is called venomous (as opposed to a species that is poisonous upon ingestion!). In general though, I would recommend rephrasing this to make this a bit more scientifically accurate. The genus Odontomachus not just “comprises” venomous ants, but ALL members of this genus are venomous due to the presence of a stinger, which in fact is a character all members of the subfamily Ponerinae possess. I recommend rephrasing this accordingly.
- Sample size
I am a bit concerned about the seemingly small sample sizes used in this study, especially since the results are meant to paint a very general picture, i.e. species discrimination between these morphologically closely related species which is in and of itself a useful and relevant tool.
Secondly, all specimens have been collected in Thailand and I can’t tell from Table 1 if the specimens were part of nest series i.e. if collections from the same locations represent two individuals from the same nest or not. Furthermore, personally, I would like to see some notes from the specific collection site (in log, on ground, etc.) and method (hand collecting, Winkler extraction etc.) – this information could also be presented in a more detailed table as part of an appendix.
A solution to the geographical limitation of the study would be to mention this explicitly in the title, for instance: “Species discrimination of three Odontomachus (Formicidae: Ponerinae) species in Thailand using outline morphometrics”
Another solution would obviously be to include more specimens, especially specimens from other locations within the species ranges. I realize that making new collections is likely not possible due to time and money constraints, yet I wonder if existing digital images could be used to expand sample size.
My concerns about sample size are about capturing the true variation of the two characters that have been used in the study (pronotum outline and head outline), this issue also becomes apparent when looking at the large error bars in Figure 4…
- Use of mandibles as a character
I am a bit surprised and curious about the authors’ choice not to use mandibles as a third body part. There is obviously variation in mandible shape between the 3 species and it appears to be a very viable character for this type of analysis. I can’t help but wonder if adding this third character would lead to a more complete picture for discriminating these species.
- Viewing/measurement angle
When looking at Figure 5, I wonder about the importance of viewing angle when making these measurements. I realize, the authors address this in Line 107, yet they do so very broadly and of little use to someone who would like to follow these methods for their own analysis. The superposition in Figure 5 (especially the lack of symmetry of the posterior part of the pronotum with a lack on the left side compared to perfect symmetry on the right side) may suggest that a slight angling could render vastly different results…
- O. latidens
I’m missing a proper mention of O. latidens and how it relates to the other species morphologically. Of course, I know that it is a much larger species and looks completely different, making IDing/distinguishing it not so difficult but I believe the readers would like to know this as well, especially since the species is mentioned at in the beginning of the paper.
Additional items:
- Line 1: The title appears very unprofessional: First of all the genus should be italicized, second, “worker ants” is not a scientific term and third, I recommend adding the proper taxonomic classification in the title in parentheses next to the genus (Formicidae: Ponerinae)
- Line 50: Is there a reference for 15.000 ant species worldwide? This is likely an accurate number but I would prefer a clear reference to it.
- Line 58: I would recommend to add the proper taxonomic classification of the genus at an earlier point, for instance as part of the title or when the word “ant” is FIRST used.
- Line 87: Saying “female specimens” is extremely misleading when talking about ants, this should say “a total of 80 workers” or “a total of 80 specimens” (as the caste is mentioned a few sentences later anyways)
- Line 102: “like non-social insects” – I do not know what is meant by that…
In summary, I do support publication of this paper after the points I mentioned have been adequately addressed.
Reviewer 3 Report
2 February 2022, Referee's report for Insects on
Yudthana Samung, Tanawat Chaiphongpachara, Jiraporn Ruangsittichai, Patchara Sriwichai, Anon Phayakkaphon, Weeyawat Jaitrong, Jean-Pierre Dujardin and Suchada Sumruayphol:
Species discrimination of three Odontomachus worker ants using outline morphometrics
Decision
The manuscript cannot be accepted in the present form. Re-submission is encouraged as I see some chance to achieve a more convincing presentation after the data have been reanalyzed using exploratory data analyses and discriminant functions that avoided data overfitting. For details see comments below and my comments in the manuscript file.
Comments
The authors describe an outline-based classification approach (OM) to distinguish three Odontomachus species from Indochina. They have chosen three easily separable species the workers of which differ in surface sculpture, pubescence density, pilosity distribution, mesosoma and head shape and mandibular dentition: Odontomachus monticola, O. rixosus and O. simillimus. The species furthermore show relatively little size variation, weak allometries and no caste polymorphism. All these factors should facilitate the success of an applied classification method even if it has no strong discriminative power.
The authors stated on line 143 that the first 21 components were used for the discriminant model and subsequent validated reclassification. This is major systematic mistake. It is an iron law that the number of variables introduced into a linear discriminant analysis should at least be threefold the number of specimens in the smallest class. If there are, as in this study, the number of variables 21 but the number of individuals is only 22 in O.simillimus, then the LDA behaves like a compliant servant of the researcher to confirm his/her possibly wrong hypotheses. We have here extreme overfitting. In other words the authors have either to increase the sample size that the rarest species achieves 66 individuals or to reduce the number of used variables to 7. To make the study persuasive, they should calculate the analyses anew. First they, should reduce the number of PCA components to seven. I recommend doing this character reduction by forward or backward selection with a discriminant analysis. This is the first reasonable step which, however, still has the disadvantage of being a hypothesis-driven analysis. Then a hard check of the established hypothesis by an exploratory data analysis (EDA) should follow. I recommend different hierarchical analyses such as UPGMA, Ward or k-means clustering, NMDS etc. In the EDAs, character reduction is not essential as there is no hypothesis imposed – you might well use 21 PCA components or a reduced data set. The best result can be found by trial and error. Alternately, you may start the whole classification procedure with an EDA that generates a species hypothesis and then control this hypothesis with a character-reduced discriminant analysis. I write this because the authors did not explain how the initial species hypotheses were established. By subjective assessment of the overall phenotype?
An good example of how a stepwise classification procedure for extremely cryptic ants species by interaction between exploratory and hypothesis-driven can be done is given by Seifert et al. (2017). Seifert B, Okita I, Heinze J 2017: A taxonomic revision of the Cardiocondyla nuda group (Hymenoptera: Formicidae). – Zootaxa 4290 (2):324-356.
The authors missed a published study where the performance to distinguish similar Myrmica species was directly compared between a combined GM+OM-approach and a conventional linear morphometrics approach:
Bagherian, A., Münch, W., Seifert, B. (2012): A first demonstration of interspecific hybridization in Myrmica ants by geometric morphometrics (Hymenoptera: Formicidae) – Myrmecological News 17: 121-131.
They furthermore missed a paper directly comparing the performance of GM+OM against conventional linear morphometrics in truly cryptic (almost inseparable) taxa:
Seifert, B., Bagherian Yazdi, A., Schultz, R. (2014): Myrmica martini sp.n. – a cryptic species of the Myrmica scabrinodis species complex (Hymenoptera: Formicidae) revealed by geometric morphometrics and nest-centroid clustering. – Myrmecological News 19: 171-181.
The tree in Fig. 7 gives only the data of the collected samples of the species. The crucial question is here if a hierarchical tree (or another exploratory data analysis) can separate the species based on worker individuals with a minimum error This would then provide direct evidence for the truth of subjectively pre-established species hypotheses. This can be tried by a variant of NC clustering.
See
Seifert, B., Ritz, M., Csösz, S. (2013): Application of Exploratory Data Analyses opens a new perspective in morphology-based alpha-taxonomy of eusocial organisms. –Myrmecological News 19: 1-15.
or
CsÅ‘sz, S. & Fisher, B.L. (2015) Diagnostic survey of Malagasy Nesomyrmex species-groups and revision of hafahafa group species via morphology based cluster delimitation protocol. ZooKeys, 526, 19–59. https://doi.org/10.3897/zookeys.526.6037

Round 2
Reviewer 1 Report
After read the response letters from authors to the three reviewers and look at the edited version of the revised manuscript, I find that all questions were answered. We have now a much better written MS, also in terms of data analysis and clarity of results.
Minor corrections needed:
Line 71: insert blank space (‘reported from’);
Line 139: insert blank space (‘1 minus R’);
Line 425: the reference Da Silva Camargo et al. should be placed in a new line.